# Development of transgenic models susceptible and resistant to SARS-CoV-2 infection in FVB background mice

**Sun-Min Seo**[1], **Jae Hyung Son**[2], **Ji-Hun Lee**[1], **Na-Won Kim**[1], **Eun-Seon Yoo**[1], **Ah-Reum Kang**[1], **Ji Yun Jang**[2,3], **Da In On**[4], **Hyun Ah Noh**[4], **Jun-Won Yun**[5,6], **Jun Won Park**[7], **Kang-Seuk Choi**[8], **Ho-Young Lee**[9], **Jeon-Soo Shin**[10], **Jun-Young Seo**[11], **Ki Taek Nam**[11], **Ho Lee**[2☯]*, **Je Kyung Seong**[12☯]*, **Yang-Kyu Choi**[1☯]*

**1** Department of Laboratory Animal Medicine, College of Veterinary Medicine, Konkuk University, Seoul, Republic of Korea, **2** Graduate School of Cancer Science and Policy, National Cancer Center, Goyang, Gyeonggi, Republic of Korea, **3** College of Pharmacy, Dongguk University, Seoul, Republic of Korea, **4** Korea Mouse Phenotyping Center (KMPC), Seoul National University, Seoul, Republic of Korea, **5** Department of Medical and Biological Sciences, The Catholic University of Korea, Bucheon, Republic of Korea, **6** Laboratory of Veterinary Toxicology, College of Veterinary Medicine, Seoul National University, Seoul, Republic of Korea, **7** Division of Biomedical Convergence, College of Biomedical Science, Kangwon National University, Chuncheon, Republic of Korea, **8** Laboratory of Avian Diseases, College of Veterinary Medicine, Seoul National University, Seoul, Republic of Korea, **9** Department of Nuclear Medicine, Seoul National University Bundang Hospital, Seongnam, Republic of Korea, **10** Department of Microbiology, Brain Korea 21 Project for Medical Science, Yonsei University College of Medicine, Seoul, Republic of Korea, **11** Severance Biomedical Science Institute, Brain Korea 21 Project for Medical Science, Yonsei University College of Medicine, Seoul, Republic of Korea, **12** Laboratory of Developmental Biology and Genomics, Research Institute for Veterinary Science, and BK 21 PLUS Program for Creative Veterinary Science Research, College of Veterinary Medicine, Seoul National University, Seoul, Republic of Korea

☯ These authors contributed equally to this work.
* ho25lee@ncc.re.kr (HL); snumouse@snu.ac.kr (JKS); yangkyc@konkuk.ac.kr (YKC)

**Data Availability Statement:** All relevant data are within the paper and its Supporting information files.

## Abstract

Coronavirus disease (COVID-19), caused by Severe Acute Respiratory Syndrome Coronavirus 2 (SARS-CoV-2), is currently spreading globally. To overcome the COVID-19 pandemic, preclinical evaluations of vaccines and therapeutics using K18-hACE2 and CAG-hACE2 transgenic mice are ongoing. However, a comparative study on SARS-CoV-2 infection between K18-hACE2 and CAG-hACE2 mice has not been published. In this study, we compared the susceptibility and resistance to SARS-CoV-2 infection between two strains of transgenic mice, which were generated in FVB background mice. K18-hACE2 mice exhibited severe weight loss with definitive lethality, but CAG-hACE2 mice survived; and differences were observed in the lung, spleen, cerebrum, cerebellum, and small intestine. A higher viral titer was detected in the lungs, cerebrums, and cerebellums of K18-hACE2 mice than in the lungs of CAG-hACE2 mice. Severe pneumonia was observed in histopathological findings in K18-hACE2, and mild pneumonia was observed in CAG-hACE2. Atrophy of the splenic white pulp and reduction of spleen weight was observed, and hyperplasia of goblet cells with villi atrophy of the small intestine was observed in K18-hACE2 mice compared to CAG-hACE2 mice. These results indicate that K18-hACE2 mice are relatively susceptible to SARS-CoV-2 and that CAG-hACE2 mice are resistant to SARS-CoV-2. Based on these lineage-specific sensitivities, we suggest that K18-hACE2 mouse is suitable for highly

**Funding:** This work was supported by grants for building an infrastructure to support preclinical trials (2021M3H9A1030260) and KMPC Grants (2014M3A9D5A01075128 and 2020M3A9D5A01082439) from the Korea Ministry of Science, Technology, and ICT.

**Competing interests:** The authors have declared that no competing interests exist.

susceptible model of SARS-CoV-2, and CAG-hACE2 mouse is suitable for mild susceptible model of SARS-CoV-2 infection.

## Introduction

Severe acute respiratory syndrome coronavirus 2 (SARS-CoV-2), first reported in December 2019, is a highly contagious pathogen. SARS-CoV-2, which belongs to the genus Betacorona-virus, is the causative agent of the COVID-19 pandemic [1, 2]. Symptoms of COVID-19 include fever, dry cough, fatigue, body aches, sore throat, and loss of taste or smell [3, 4]. According to the World Health Organization's situation report, from the date of the first report to December 2021, the cumulative number of reported cases and deaths was 273 million and 5.3 million, respectively [5].

To respond to the rapid spread of SARS-CoV-2, basic research on SARS-CoV-2, research and development, and preclinical evaluations of vaccines and therapeutics are being actively conducted. To support these studies, COVID-19 preclinical animal models are essential. Numerous animal models, including Syrian hamster [6–8], ferret [9–11], and several species of non-human primates [12, 13], have been used in preclinical trials and research on SARS-CoV-2 since the severe acute respiratory syndrome (SARS) outbreak in 2002. However, these animal models cannot reproduce equivalent clinical symptoms, pathological changes, or mortality observed in COVID-19 patients [14].

Transgenic mice exhibit the most similar clinical symptoms and pathological changes to SARS and COVID-19 as those seen in humans [15–21]. In particular, transgenic mice geneti-cally engineered to express human angiotensin-converting enzyme 2 (hACE2), a functional receptor of SARS-CoV and SARS-CoV-2 [22], are the most commonly used. Due to low SARS-CoV-2 susceptibility, wild-type mice are unsuitable for SARS-CoV-2 study [23]. How-ever, hACE2 transgenic mice exhibit mild to severe pathological lung changes with clinical symptoms similar to those of COVID-19 patients. K18-hACE2 [16, 19, 21] and CAG-hACE2 [15, 18, 20] are commonly used hACE2 transgenic mice.

Although many studies have used K18-hACE2 and CAG-hACE2 transgenic mice, a com-parative study of SARS-CoV-2 infection in these transgenic mouse lineages has not been pub-lished. In this study, we infected an equal amount of SARS-CoV-2 to two different lineages of FVB background mice, K18-hACE2 and CAG-hACE2. Here, we present detailed data on the distribution of SARS-CoV-2, clinical manifestations, and histopathology to compare SARS-CoV-2 susceptibility and resistance between two different lineages of transgenic mice driven by different promoters.

## Materials & methods

### 1. Generation of hACE2 transgenic mice

Two different lineages of human ACE2 gene transgenic mice, FVB-Tg (K18-ACE2)K [K18-hACE2] and FVB-Tg (CAG-ACE2) [CAG-hACE2], were produced and provided by the National Cancer Center of the Republic of Korea. The human ACE2 gene (HG10108-UT, Sino Biological Inc., China) was cloned behind the Krt18 (#44580, Addgene, USA) or CAGGs (#127346, Addgene) promoter. Pronuclear injections to generate transgenic mice were per-formed as previously described [24]. Two founders were used to identify clones in which the hACE2 protein was expressed in lung tissue. Human ACE2 expression was detected by

**Table 1. PCR primers for identifying hACE2 transgenic mouse.**

| Gene | | Sequence | PCR product size |
|---|---|---|---|
| **CAG-hACE2** | Forward | `5'- CGCAGCCATTGCCTTTTATGG- 3'` | 609 bp |
| | Reverse | `5'- CCAGCATTATTCATGTTTTGG- 3'` | |
| **K18-hACE2** | Forward | `5'- CACTCTGCGATATAACTCGGG- 3'` | 295 bp |
| | Reverse | `5'- CCAGCATTATTCATGTTTTGG -3'` | |

western blotting or immunohistochemistry and PCR using an anti-hACE2 antibody (#108209, Abcam, UK) and hACE2 specific primers (Table 1), respectively.

## 2. SARS-CoV-2 virus infection

The original viral stock of SARS-CoV-2 (NCCP 43326) was obtained from the National Culture Collection for Pathogens (NCCP), which is managed by the Korea Disease Control and Prevention Agency (KDCA). The viral stock was amplified by passaging in Vero E6 cells (CRL 1586, ATCC, USA), and the viral titer was confirmed using the $TCID_{50}$ assay. Amplified stocks were diluted and aliquoted to working stocks of the virus ($1.0 \times 10^2$ $TCID_{50}$/20 μl and $1.0 \times 10^5$ $TCID_{50}$/20 μl). All experiments involving infectious SARS-CoV-2, including animal infection, were handled at the biosafety level 3 facility at Konkuk University by trained researchers. All procedures were approved conducted under the guidelines of the Institutional Biosafety Committee of Konkuk University (KUIBC-2021-06) and the Konkuk University Animal Care and Use Committee (KU20142-4).

## 3. Viral infection of two different lineage mice

Two different lineages of male mice, ranging from 9 to 12 weeks of age, were grouped into a high dose (VH group), low dose of virus group (VL group), and negative control group (NC group). Mice were anesthetized with a Zoletil-xylazine cocktail and infected with 20ul of the indicated SARS-CoV-2 virus stock via intranasal route. After infection of SARS-CoV-2 virus, environmental enrichment including wood chew block and pulp house (Woojung Bio, Republic of Korea) were provided to diminish distress. The infected mice were observed daily for signs of illness and weighed to assess the progression of SARS-CoV-2 infection. In addition, illness severity was evaluated using the activity score and survival rate. The activity score was independently classified based on the following criteria: 100%, healthy; 75%, ruffled fur but active; 50%, ruffled fur, lethargy, and hunched posture; 25%, moribund; 0%, dead. Infected mice were sacrificed by exsanguination at 1, 2, 5, and 7-days post-infection (dpi). Humane euthanasia performed on the day reaching 25% of activity score. Mice dead before reaching the humane euthanasia criteria were immediately necropsied after confirmation of death. These procedures were performed under 2.5% isoflurane anesthesia using an isoflurane vaporizer (VetEquip, USA). After euthanasia, organs, including the lung, spleen, liver, kidney, heart, and small intestine, were harvested. The weights of the lungs, spleen, liver, kidneys, and heart were measured to compare the organ-to-body weight ratio. Lung tissue specimens were isolated and frozen at -80°C.

## 4. Standard curve of SARS-Cov-2

Viral RNA was isolated from the SARS-CoV-2 virus stock, which was quantified with the $TCID_{50}$ assay, using the MagNA Pure 96 system (Roche, Switzerland). cDNA was synthesized from eluted viral RNA samples using Maxime™ RT PreMix (Random Primer, iNtRon Biotechnology, Republic of Korea), followed by serial dilution. Each diluted cDNA sample (1 μl) was

used for quantitative real-time PCR (qRT-PCR). qRT-PCR was conducted using AccuPower® 2X GreenStar™ qPCR Master Mix (Bioneer, Republic of Korea) with primers targeting the ORF1ab gene of SARS-CoV-2 virus. (ORF1ab-F: 5′-CCCTGTGGGTTTTACACTTAAAAA-3′; ORF1ab-R: 5′-GATTGTGCATCAGCTGACTG -3′). The cycling procedure was as follows: pre-denaturation at 95°C for 3 min, denaturation at 95°C for 10 s, and annealing/extension at 51°C for 30 s. A total of 40 cycles were performed using a CFX96 Touch Real-Time PCR Detection System (Bio-Rad, USA). The standard curve and equation were derived based on the $TCID_{50}$ values and quantification cycles (cq) of each serially diluted sample.

## 5. Virus titer in the tissue specimens of infected mice

Lung, cerebrum, cerebellum, and duodenum specimens of infected and uninfected mice were manually emulsified with phosphate-buffered saline (Bioneer) using a 3-way stop-cock and syringe. SARS-CoV-2 in specimens was inactivated using MagNA Pure 96 External Lysis buffer (Roche) after emulsification. After inactivation of the virus, viral RNA was isolated from each sample using a MagNA Pure 96 system (Roche). cDNA synthesis and real-time PCR were performed as described above. The viral load of each mouse was calculated using the cq value of each sample. Based on the standard curve, below limit of detection ($1.0 \times 10^2$ $TCID_{50}$/sample) considered as zero.

## 6. Western blot

Tissues were lysed in RIPA buffer (R4100-010, Gendepot, Barker, USA) supplemented with protease inhibitor cocktail (P3300-001, Gendepot) on ice and homogenized with stainless steel beads (Qiagen, USA) in TissueLyser II (Qiagen). The lysates were added to sample loading dye NuPAGE™ LDS sample buffer, heated to 95°C for 10 min and were separated on precast polyacrylamide gels (Bolt 4–12% Bis-Tris Plus, Thermo Fisher Scientific Inc., USA). After separation, proteins were transferred to a nitrocellulose membrane. After blocking the nonspecific site with 3% bovine serum albumin (BSA), the membrane was then incubated with anti-hACE2 antibody (1:2000, ab108209, Abcam) in 3% BSA at 4°C for overnight. The membrane was further incubated for 1 h with a peroxidase-conjugated secondary antibody (1:5000, Santa Cruz, USA) at room temperature. Immunoactivity proteins were detected with WESTSAVE Gold detection reagents (LF-QC0103, AbFrontier, Republic of Korea).

## 7. Histopathological evaluation

Organs, including the lungs, spleen, and small intestine, were harvested from euthanized mice and fixed with 10% neutral buffered formalin. After routine processing and paraffin embedding, sections were stained with hematoxylin and eosin. The sections were examined under a BX51 light microscope (Olympus, Japan), and images of each section were captured using DP74 software (Olympus). The histopathological severity of each organ was scored according to the criteria listed in Table 2.

## 8. Immunohistochemistry

After deparaffinization in xylene and rehydration in graded alcohol, the tissue sections were quenched with 3% $H_2O_2$ for 30 min and incubated in 10 mM citrate buffer solution (pH 6.0) at 90°C for 10 min for antigen retrieval. After washing with PBS, the sections were incubated with anti-hACE2 antibody (1:2000, ab108209, Abcam) at 4°C for overnight. The sections incubated for 1h at room temperature with biotinylated goat anti-rabbit secondary antibody (Vector Laboratories, USA), followed by incubation with an avidin-biotin complex (Vector

**Table 2. Histopathological scoring for quantify the states of the lung and small intestine.**

| Parameter | | 0 | 1 | 2 | 3 | 4 | 5 |
|---|---|---|---|---|---|---|---|
| Lung scoring | Pneumonia | None | < 10% of entire lung | < 25% of entire lung | < 50% of entire lung | < 75% of entire lung | ≥ 75% of entire lung |
| | Perivascular edema | None | Mild | Moderate | Severe | | |
| | Bronchitis and bronchiolitis | None | Mild | Moderate | Severe | | |
| Small intestine scoring | Goblet cell hyperplasia | Average of goblet cells from at least 10 villi of jejunum. | | | | | |

Laboratories) for 1h at room temperature and stained with DAB Chromogen (ScyTek Laboratory, West Logan, USA) for 5 min at room temperature, washed with PBS and deionized water and counterstained with hematoxylin.

## 9. Statistical analysis

Body weight, organ weight, survival analysis, histopathological scoring, and pulmonary viral titer were compared between the different groups of each transgenic lineage. Unpaired two-tailed t-tests and log-rank tests were used to calculate p-values using GraphPad Prism 7.04 (GraphPad Software, USA). Statistical significance was set at $P < 0.05$.

## Results

### 1. Different characterization between K18-hACE2 and CAG-hACE2 mice

hACE2 was cloned using Krt18 and CAGGs promoters, and pronuclear injection was performed to obtain K18-hACE2 and CAG-hACE2 transgenic lineages. Western blotting and immunohistochemistry were performed to confirm protein levels of hACE2 in each transgenic lineage (Fig 1). In CAG-hACE2, the expression level of hACE2 protein in the kidney, lung,

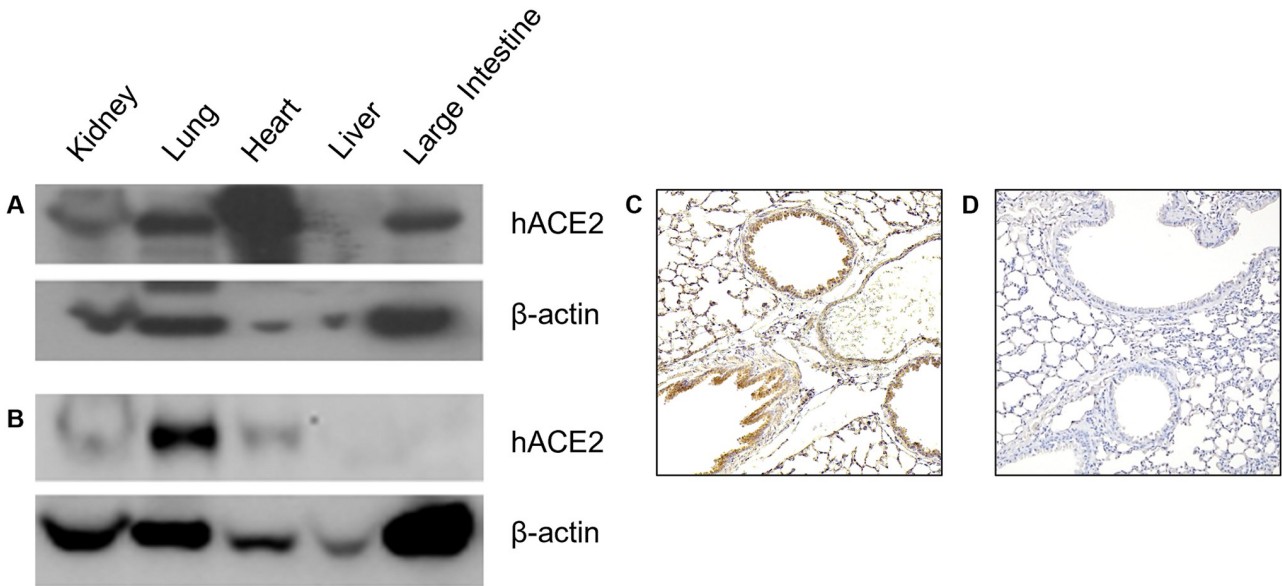

**Fig 1. hACE2 expression profile in multiple organs of K18-hACE2 and CAG-hACE2 transgenic lineage.** Elevated levels of hACE2 were detected in the lungs of both K18-hACE2 and CAG-hACE2 transgenic lineages. (A) Western blot analysis of hACE2 expression in multiple organs of CAG-hACE2; (B) Western blot analysis of hACE2 expression in multiple organs of K18-hACE2; (C) Immunohistochemistry analysis of hACE2 expression in the lungs of K18-hACE2; (D) Immunohistochemistry analysis of hACE2 expression in the lungs of mock control.

heart, and large intestine was similar to or higher than that of beta-actin, a housekeeping gene (Fig 1A). In K18-hACE2, the expression level of hACE2 protein in the lung was higher than that of beta-actin, but the rest of the organs were lower or undetectable (Fig 1B). The expression of hACE2 in lung tissue was confirmed by immunochemistry. hACE2 protein was expressed at high levels in the respiratory epithelium (Fig 1C).

## 2. Morbidity and mortality difference induced by SARS-CoV-2 in K18-hACE2 and CAG-hACE2 mice

To minimize factors that could affect SARS-CoV-2 infection, 40 and 33 age-matched K18-hACE2 and CAG-hACE2 transgenic mice were grouped into the high dose virus group (VH), low dose virus group (VL), and negative control group (NC). These groups of mice were inoculated with working stocks of SARS-CoV-2 ($1.0 \times 10^2$ $TCID_{50}$/20 μl and $1.0 \times 10^5$ $TCID_{50}$/20 μl) via the intranasal route. Morbidity and mortality of the infected mice were observed daily. Four to five K18-hACE2 mice in the VL and VH groups and three to four CAG-hACE2 mice in the VL and VH groups were sacrificed at 1, 2, 5, and 7 dpi. However, all K18-hACE2 mice in the VH group died at 6 dpi and were necropsied (n = 3). The NC group of each lineage of mice was sacrificed on the final day of the study at 7 dpi.

As shown in Fig 2, the body weight of VH group of K18-hACE2 decreased by 22% at 6 dpi compared to the initial body weight. The VL group of K18-hACE2 exhibit minor body weight recovery at 7 dpi, after an 11% decrease at 6 dpi. At the endpoint, the body weight of both the VH and VL groups of K18-hACE2 was significantly different from that of the NC group (Fig 2A) (P < 0.01). However, both the VH and VL groups of CAG-hACE2 exhibited only a slight decrease compared with the initial body weight (5.9% in the VH group and 1.5% in the VL group). The VL group of CAG-hACE2 mice gradually regained weight after 3 dpi. At 7 dpi, the body weight of both the VH and VL groups of CAG-hACE2 significantly differed from that of the NC group (Fig 2B) (P < 0.001, P < 0.01). The activity score of the K18-hACE2 VH group continued to decrease after the first decline at 4 dpi, exhibiting moribund traits at 5 dpi. The activity score of the K18-hACE2 VL group started to decline from 4 dpi, dropping to 64% at 7 dpi. The VH group of the K18-hACE2 exhibited significant differences compared to the NC and VL groups (Fig 2C) (P < 0.05). The CAG-hACE2 lineage, in marked contrast to the K18-hACE2 lineage, did not exhibit clinical manifestations until 7 dpi (Fig 2D). The VH group of K18-hACE2 showed significantly lower activity score than VH group of CAG-hACE2 (P<0.05). Death of the VH and VL groups of K18-hACE2 began at 5 dpi (n = 5 in the VH group and n = 1 in the VL group), with the VH and VL groups exhibiting 100% and 64% mortality by 6 dpi (n = 3 in the VH group and n = 1 in the VL group) and 7 dpi (n = 2 in the VL group), respectively. Mortality in the VH group significant differed in the log-rank test compared to that in the NC and VL groups (Fig 2E) (P < 0.01). Similar to the activity score, VH and VL groups of CAG-hACE2 mice did not die during the entire study period (Fig 2F). Mortality rate in VL and VH groups were significantly higher in K18-hACE2 than in CAG-hACE2 (P < 0.05, P < 0.001).

## 3. Kinetics of SARS-CoV-2 replication

Based on the remarkable differences in clinical manifestations between K18-hACE2 and CAG-hACE2, we compared the pulmonary, cerebral, cerebellar, and intestinal distribution of SARS-CoV-2 and the virus kinetics of the two different lineages (Fig 3). Pulmonary viral replication in the VL group of K18-hACE2 (Fig 3A) and CAG-hACE2 (Fig 3B) reached maxima at 7 and 5 dpi, respectively, with median values of $3.9 \times 10^4$ and $1.6 \times 10^4$ $TCID_{50}$ SARS-CoV-2/lung, respectively. Virus titers in the VH group of K18-hACE2 and CAG-hACE2 reached

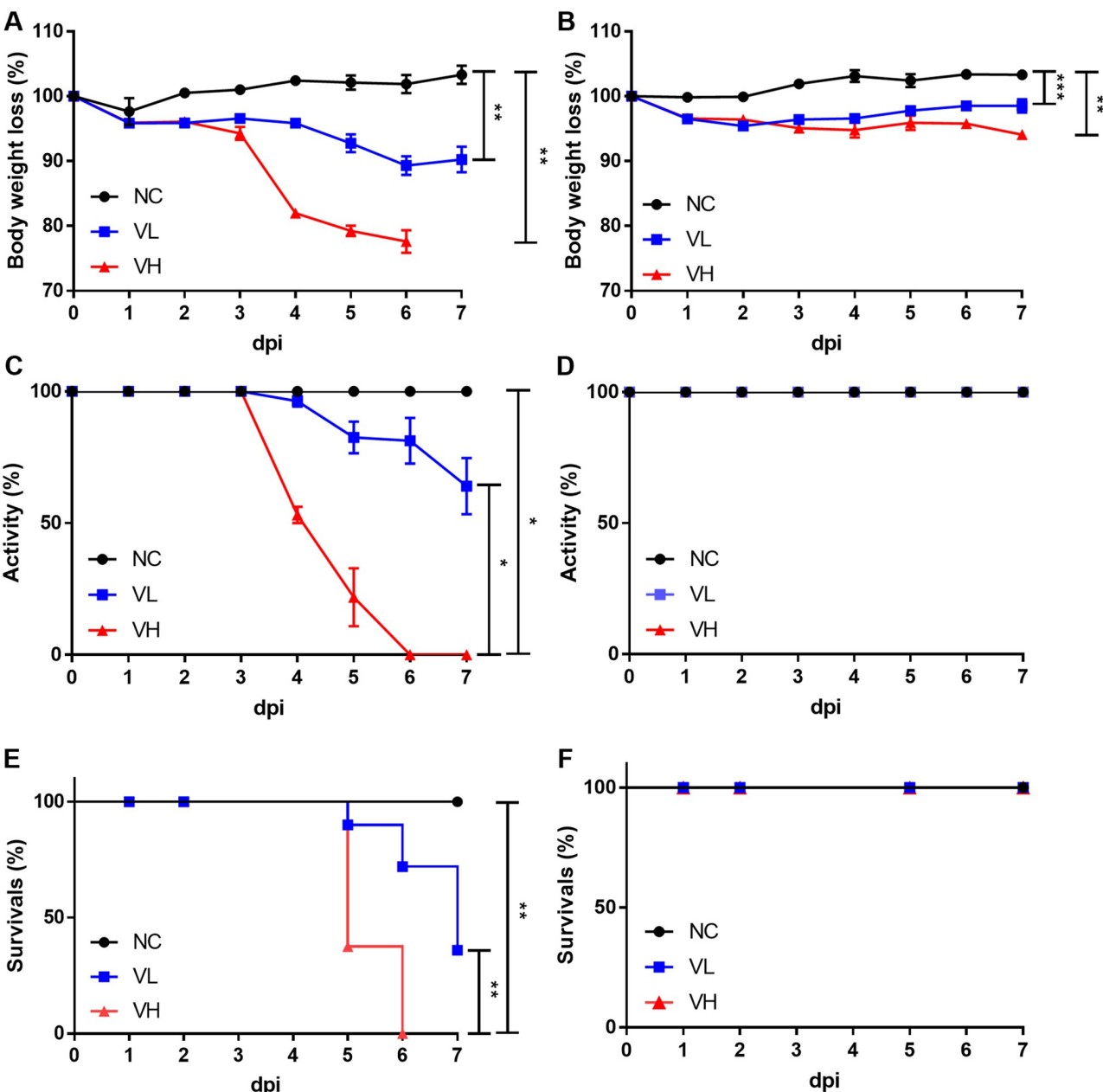

**Fig 2. Differences in morbidity and mortality due to SARS-CoV-2 infection in K18-hACE2 (n = 40) and CAG-hACE2 (n = 33) lineages.** (A) shows weight loss of K18-hACE2; (B) shows weight loss of CAG-hACE2; (C) shows activity score of K18-hACE2; (D) shows activity score of CAG-hACE2. (E) shows survival rate of K18-hACE2; (F) shows survival rate of CAG-hACE2. Data is represented as mean ± SE at indicated day post-infection (dpi). *P < 0.05; **P < 0.01; ***P < 0.001 as assessed using unpaired two-tailed t-test and log-rank test.

maximum titers at 5 dpi and 2 dpi, respectively, with median values of $1.1 \times 10^7$ and $8.6 \times 10^5$ $TCID_{50}$ SARS-CoV-2/lung. Viral titer gradually decreased after reaching the maxima. During the entire study, in VL groups of both lineages, the titers of SARS-CoV-2 were approximately $10^4$ $TCID_{50}$ SARS-CoV-2/lung. However, in the VH group, the viral titer of CAG-hACE2 appeared to be $10^5$–$10^6$ $TCID_{50}$ SARS-CoV-2/lung, whereas K18-hACE2 appeared to be relatively high at $10^6$–$10^7$ $TCID_{50}$ SARS-CoV-2/lung. Comparing the VL and VH groups in each lineage, a significant difference was observed in the virus titer of each group at 2 and 7 dpi for

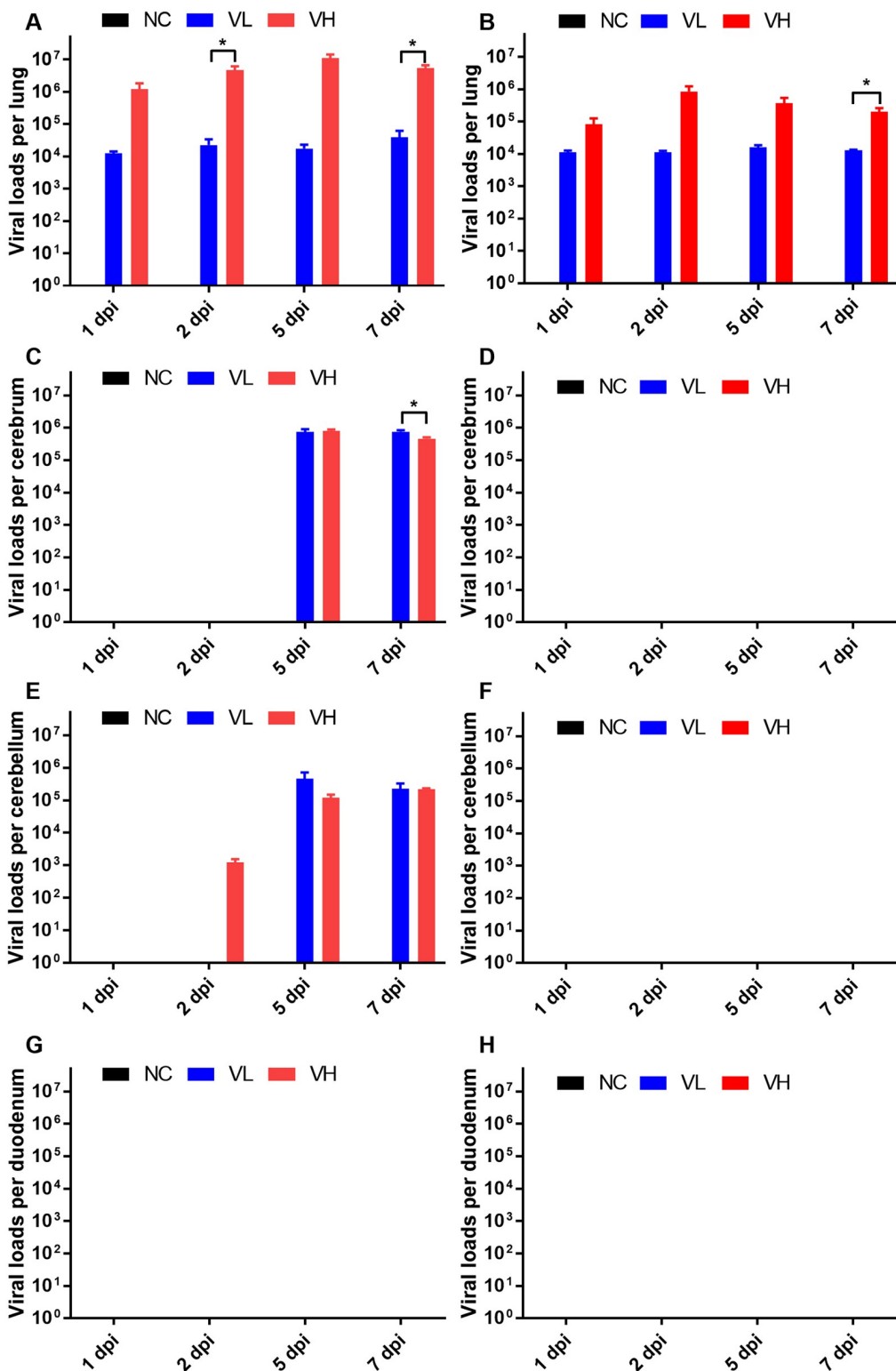

**Fig 3. Kinetics of SARS-CoV-2 replication in organs of K18-hACE2 and CAG-hACE2 transgenic mice.** Kinetics of SARS-CoV-2 replication in lungs of K18-hACE2 (A) and CAG-hACE2 (B); kinetics of SARS-CoV-2 replication in cerebrum of K18-hACE2 (C) and CAG-hACE2 (D); kinetics of SARS-CoV-2 replication in cerebellum of K18-hACE2 (E) and CAG-hACE2 (F); kinetics of SARS-CoV-2 replication in duodenum of K18-hACE2 (G) and CAG-hACE2 (H). Titers of SARS-CoV-2 were measured with quantitative PCR and expressed as TCID$_{50}$ SARS-CoV-2/organ. Data are

represented as mean ± SE at indicated day post-infection (dpi). The VH group of K18-hACE2 transgenic mice died at 6 dpi, marked as 7 dpi. *P < 0.05 as assessed using unpaired two-tailed t-test.

K18-hACE2 and 7 dpi for CAG-hACE2 (P < 0.05). At 2dpi, the VH group of K18-hACE showed significantly higher pulmonary viral replication than VH group of CAG-hACE2. (P < 0.05).

Viral replication in the cerebrum of K18-hACE2 appeared at 5 dpi and 7 dpi for both the VL and VH groups (Fig 3C). A significant difference appeared between the VL and VH groups at 7 dpi (P < 0.05). Contrary to the results of K18-hACE2, in CAG-hACE2, viral replication was not detected in the cerebrum of both the VL and VH groups (Fig 3D). Comparing between K18-hACE2 and CAG-hACE2, K18-hACE2 showed significantly higher viral replication in 5 and 7 dpi of VL (P <0.05, P <0.001) and VH groups (P < 0.001, P < 0.05).

Viral replication in the cerebellum of K18-hACE2 was slightly lower than that in the cerebrum (Fig 3E). As in the cerebrum of CAG-hACE2, no viral replication was detected in the cerebellum of CAG-hACE2 (Fig 3F). Comparing the viral replication of the cerebellum of K18-hACE2 and CAG-hACE2, the VH group of K18-hACE2 showed significantly higher viral replication at 2, 5, and 7 dpi (P < 0.01, P < 0.05, P < 0.01). And viral replication of the small intestine was inspected. However, no viral replication was observed in both the VL and VH groups of K18-hACE2 (Fig 3G) and CAG-hACE2 (Fig 3H).

## 4. Organ to body weight ratio

To compare the lineage-dependent difference in sensitivity to SARS-CoV-2, we compared the average value of organ weight to the body weight of necropsied mice on each dpi. The lungs (Fig 4A and 4B) and spleen (Fig 4C and 4D) of infected mice from the VH and VL groups

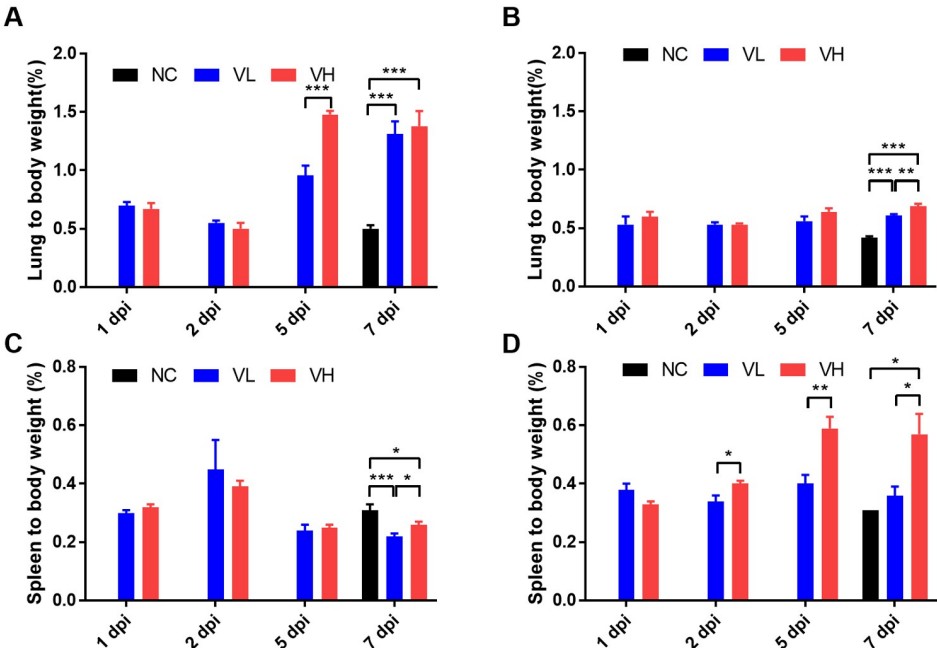

**Fig 4. Organ to body weight ratio of SARS-CoV-2 infected K18-hACE2 and CAG-hACE2 transgenic mice.** Relative weight of lungs of K18-hACE2 (A) and CAG-hACE2 (B); relative weight of spleens of K18-hACE2 (C) and CAG-hACE2 (D). Data are represented as mean ± SE at indicated day post-infection (dpi). The VH group of K18-hACE2 transgenic mice died at 6 dpi, marked as 7 dpi. *P < 0.05; **P < 0.01; ***P < 0.001 as assessed using unpaired two-tailed t-test.

were compared for both lineages. Significant differences were observed at 5 dpi and 7 dpi in the lungs of the K18-hACE2 lineage. At 5 dpi, the lung-to-body weight ratio of the VH group of the K18-hACE2 was 53% higher than that of the VL group (P < 0.001). At 7 dpi, the relative lung ratios of the VH and VL groups of K18-hACE2 were 173.6% and 159.9% higher than those of the NC group, respectively (Fig 4A) (P < 0.001). In the CAG-hACE2 lineage, at 7 dpi, the relative lung ratios of the VH and VL groups were 65.4% and 44.5% higher than those of the NC group, respectively (P < 0.001), and the VH group was 14.5% higher than that of the VL group (Fig 4B) (P < 0.01). Comparing the lung-to-body weight ratios of the two different transgenic lineages, CAG-hACE2 was lower than K18-hACE2. Comparing between K18-hACE2 and CAG-hACE2, K18-hACE2 showed significantly higher lung-to-body weight ratio at 1, 5 and 7 dpi of VL (P <0.05, P <0.01, P <0.01) and 5 and 7 dpi of VH groups (P < 0.001, P < 0.05).

In the K18-hACE2 lineage, significant differences were observed in the spleen-to-body weight ratio between the NC, VH, and VL groups at 7 dpi. The spleen-to-body weight ratio of the NC group of K18-hACE2 was 66.5% and 76.3% higher than that of the VH and VL groups, respectively (P < 0.05, P < 0.001), and that of the VH group was 5.9% higher than that of the VL group (Fig 4C) (P < 0.05). Furthermore, significant differences were found at all dpi except at 1 dpi in the spleen-to-body weight ratio of CAG-hACE2 (P < 0.05, P < 0.01, P < 0.05). At 2, 5, and 7 dpi, the ratios of the VH group of the CAG-hACE2 were 20%, 48.8%, and 58.9% higher than those in the VL group, respectively. The ratio of the VH group of the CAG-hACE2 at 7 dpi was 81.1% higher than that in the NC group (Fig 4D) (P < 0.05). Comparing between K18-hACE2 and CAG-hACE2, K18-hACE2 showed significantly lower spleen-to-body weight ratio at 1, 5 and 7 dpi of VL (P <0.05, P <0.01, P <0.05) and 5 and 7 dpi of VH groups (P < 0.01, P < 0.05).

Comparing the lung-to-body weight ratio of the two lineages, K18-hACE2 reached its highest point at 5 and 7 dpi, and CAG-hACE2 at 7 dpi. The lung ratio of K18-hACE2 increased dramatically between 2 and 5 dpi, but that of CAG-hACE2 did not increase as much as K18-hACE2.

## 5. Histopathological scoring and analysis

Based on the differences in SARS-CoV-2 titer and organ-to-body weight ratio between K18-hACE2 and CAG-hACE2, we performed histopathological examination and scoring of the lungs, spleen, and small intestine. For histopathological evaluation of the lungs, infection severity was assessed according to the following parameters: pneumonia, perivascular edema, and bronchitis/bronchiolitis (Fig 5). Each parameter was scored and summed according to the criteria (Table 2). Histopathological analysis of the lungs of both lineages revealed the development of interstitial pneumonia in the early stage of infection, followed by alveolitis, perivascular edema, and bronchitis/bronchiolitis in the last stage (Fig 5). In the histopathological scoring analysis of the K18-hACE2 lineage, no significant difference was observed in the total lung score between the VH and VL groups. However, the total score of the lungs gradually increased in both groups (Fig 6A). In the CAG-hACE2 lineage, significant differences were observed at all dpi (1 and 7 dpi: P < 0.05; 2 and 5 dpi: P < 0.01). Unlike the K18-hACE2 lineage, which worsened over time, the VH and VL groups of CAG-hACE2 exhibited a gradual decrease from 1 dpi to 5 dpi and increased at 7 dpi (Fig 6B). Comparing K18-hACE2 and CAG-hACE2, the VH group of CAG-hACE2 showed significantly higher total lung score at 1 dpi (P <0.05). Atrophy of the white pulp in the spleen due to COVID-19 infection has been reported in a few case reports [25–27]. Therefore, we performed a histopathological evaluation of the spleen. For histopathological evaluation of the spleen, the area of white pulp was

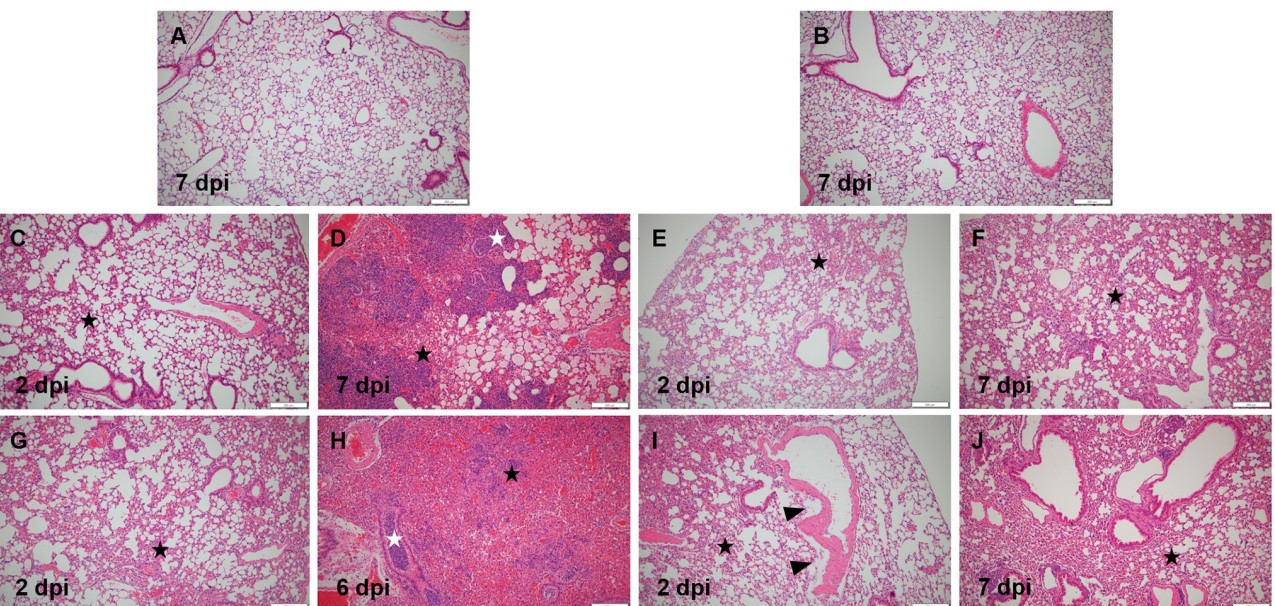

**Fig 5. Histopathological analysis of lungs of SARS-CoV-2 infected K18-hACE2 and CAG-hACE2 transgenic mice.** Lung section of a K18-hACE2 mouse of the NC group (A), VL group (C, D), VH group (G, H) and a CAG-hACE2 mouse of the NC group (B), VL group (E, F), VH group (I, J) on 2 dpi or 6 to 7 dpi. Interstitial pneumonia (black star); bronchiolitis (white star); perivascular edema (arrowhead); (magnification x100).

measured relative to the area of the entire spleen. White pulp atrophy was observed in both strains. As the severity of the disease progressed, atrophy of the splenic follicle occurred (Fig 7). A significant difference in the percentage of white pulp area between the VH and VL groups of K18-hACE2 mice at 5 dpi (P < 0.01), and the white pulp ratio of both the VH and VL groups was significantly (P < 0.001) lower than that of the NC group at 7 dpi (Fig 8A). In the CAG-hACE2 lineage, a significant difference was only observed between the VH and VL groups at 7 dpi (P < 0.05) (Fig 8B). The ratio of white pulp in the spleen gradually decreased over time in both lineages. However, it decreased more rapidly in the K18-hACE2 lineage. At 7 dpi, the white pulp ratio of the K18-hACE2 lineage was 45.5% and 51% lower in the VH and VL groups, respectively, than in the NC group; however, the white pulp ratio of the CAG-

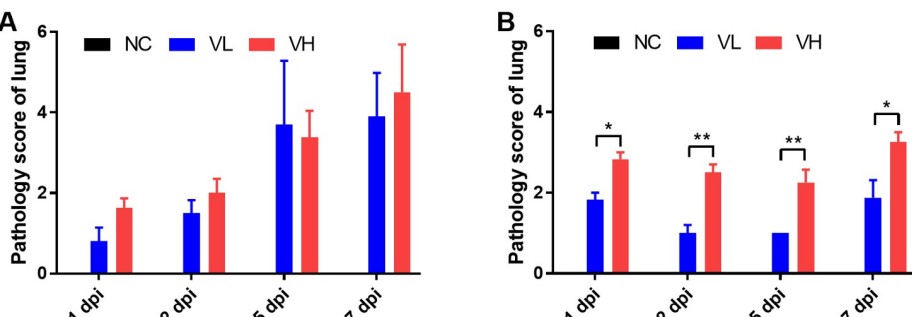

**Fig 6. Histopathological scoring of lungs of SARS-CoV-2 infected K18-hACE2 and CAG-hACE2 transgenic mice.** The lung of each group was scored as indicated in Table 2. (A) Total lung score of K18-hACE2 mice; (B) Total lung score of CAG-hACE2 mice. Data is represented as mean ± SE at indicated day post-infection (dpi). The VH group of K18-hACE2 transgenic mice died at 6 dpi, marked as 7 dpi. *P < 0.05; **P < 0.01 as assessed using unpaired two-tailed t-test.

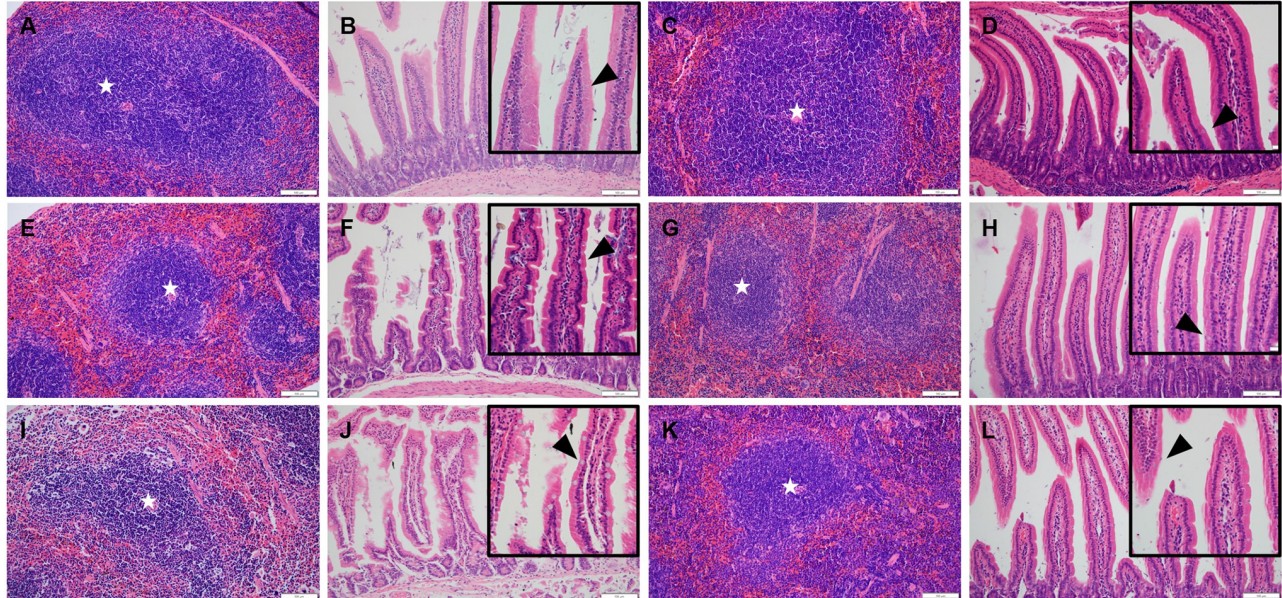

**Fig 7. Histopathological analysis of spleens, and small intestines of SARS-CoV-2 infected K18-hACE2 and CAG-hACE2 transgenic mice.** Spleen and small intestine sections of a K18-hACE2 mouse of the NC group (A, B), VL group (E, F), VH group (I, J) and a CAG-hACE2 mouse of the NC group (C, D), VL group (G, H), VH group (K, L) on 5 dpi. White pulp in spleen (star); goblet cell in small intestine (arrowhead); (magnification x200, inset x400).

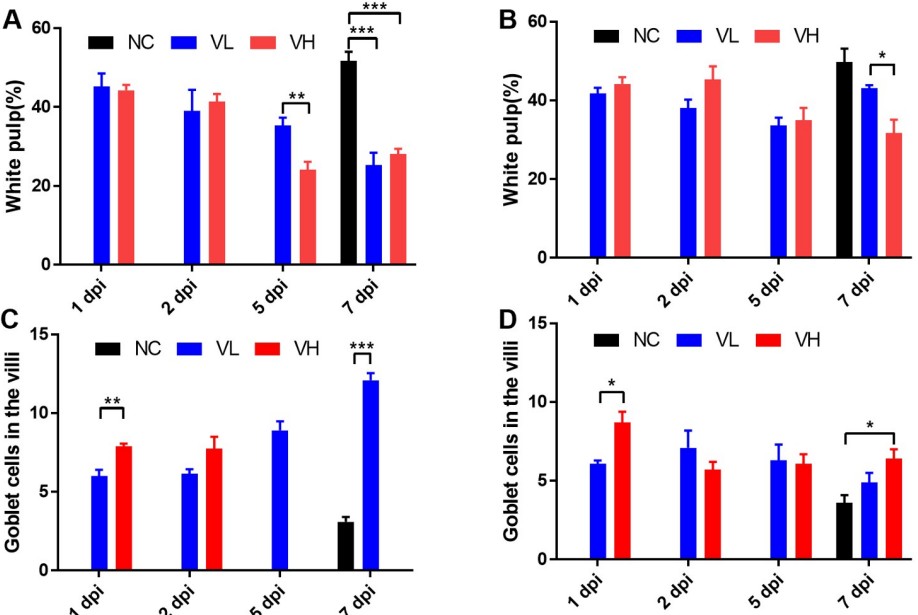

**Fig 8. Histopathological scoring of spleens and small intestines of SARS-CoV-2 infected K18-hACE2 and CAG-hACE2 transgenic mice.** Atrophy of white pulp was calculated as the area of white pulp relative to a total area of the spleen. The small intestine of each group was scored as indicated in Table 2. (A) Area of white pulp of K18-hACE2 mice; (B) Area of white pulp of CAG-hACE2 mice; (C) Number of goblet cell in small intestine of K18-hACE2 mice; (D) Number of goblet cell in small intestine of K18-hACE2 mice. Data is represented as mean ± SE at indicated day post-infection (dpi). The VH group of K18-hACE2 transgenic mice died at 6 dpi, marked as 7 dpi. *P < 0.05; **P < 0.01; ***P < 0.001 as assessed using unpaired two-tailed t-test.

hACE2 lineage was 36.4% and 13.3% lower in the VH and VL groups, respectively, than in the NC group. The white pulp ratios were significantly lower in K18-hACE2 than in CAG-hACE2 at 7dpi in VL group and 5 dpi in VH group (P<0.05). As with the severity of SARS-CoV-2 infection, hyperplasia of goblet cells occurs in the small intestine. We hypothesized that an increasing number of goblet cells indicates the severity of SARS-CoV-2 infection. Therefore, the average number of goblet cells was calculated and quantified (Table 2). Hyperplasia of the goblet cells in the villi with villi atrophy of the small intestine appeared in both lineages and was more prominent in the K18-hACE2 lineage than in the CAG-hACE2 lineage (Fig 7). The K18-hACE2 lineage exhibited significant differences between the VH and VL groups at 1 dpi (p < 0.01) and between the NC and VL groups at 7 dpi (p < 0.001). The VH group mice of the K18-hACE2 lineage, which died at 5–6 dpi, were excluded from the data due to autolysis (Fig 8C). Hyperplasia of the goblet cells and villi atrophy was not noticeable in the CAG-hACE2 lineage (Fig 8D). Comparing between K18-hACE2 and CAG-hACE2, K18-hACE2 showed a significantly higher number of goblet cells in the intestinal villi at 7 dpi of the VL group (P <0.01).

## Discussion

Ever since the SARS-CoV-2 pandemic in 2019, animal research for preclinical evaluation of therapeutics and vaccines has been ongoing worldwide. Currently, K18-hACE2 mice are primarily used for SARS-CoV-2 research and preclinical trials [28–32]. CAG-hACE2 mice are used less frequently than K18-hACE2 mice [20]. In most SARS-CoV-2 studies in mice, the transgenic lineages used for infection had a C57BL/6 background. FVB inbred strain was studied vulnerable to several viruses such as Theiler's murine encephalomyelitis virus and Hepatitis B virus than other strains, including C57BL/6 [33–35]. Considering these characteristics of the FVB strain, we expressed hACE2 in FVB background using K18 promoter and CAG promoter to infect SARS-CoV-2. The present study was conducted to compare SARS-CoV-2 susceptibility between two different hACE2 transgenic mice with an FVB background and suggests a new model for SARS-CoV-2 research. Our results suggest that susceptibility to SARS-CoV-2 infection varies depending on the promoter used for transgenic mouse production.

The clinical symptoms varied between the different transgenic lineages. K18-hACE2 exhibited a rapid decline in weight loss, survival rate, and behavior score as viral infection progressed, whereas CAG-hACE2 recovered from the disease and exhibited non-lethal clinical symptoms and resistance to infection. The clinical manifestation and lethality of K18-hACE2 that appeared in the present study correspond well with the results of an earlier study, which inoculated mice with the same viral titer, although the genetic background of the mouse was different [21]. A study investigating CAG-hACE2 in a C57BL/6 background, where mice critically infected with SARS-CoV-2, has been published [20]. However, the results regarding CAG-hACE2 mice obtained in this study were less severe than those of the previous study. These contrary results suggest the viral susceptibility may vary by the line produced even though the same vector was used. Previous study, which produced K18-hACE2 on the C57BL/6 background, showed different mortality to SARS-CoV depending on the produced line, despite the same K18 promoter being used [15].

Differences in the degree of pneumonia were observed between the two lineages. In high-titer viral infection, a higher viral load was detected in the lungs of K18-hACE2 than in CAG-hACE2. There was no difference in the viral load in the low-titer viral infection group. However, clinical symptoms observed in K18-hACE2 mice were more severe than that in CAG-hACE2 mice.

Histopathological scoring of K18-hACE2, including alveolitis, perivascular edema and bronchiolitis, was higher than that of CAG-hACE2, as in SARS-CoV-2 patients [36–39]. These results indicate that pneumonia in K18-hACE2 was more severe and K18-hACE2 mice were more susceptible to SARS-CoV-2 infection than CAG-hACE2 mice. Furthermore, CAG-hACE2 did not exhibit perivascular edema until 7 dpi, which is thought to indicate slower disease progression than K18-hACE2. The degree of viral replication in the cerebrum and cerebellum of K18-hACE2 mice was not dependent on the viral dose administered. Unlike K18-hACE2, in which the virus was detected in the cerebrum and cerebellum, the virus was not detected in CAG-hACE2. This difference seems to affect the activity score and survival rate, which decreased in K18-hACE2, whereas the CAG-hACE2 did not. This result is consistent with the previous study that virus replication in CNS affected the fatality rate [40].

Different pathological findings were observed in the spleen and the small intestine. The weight of the spleen increased at 2 dpi in all groups, except for the VL group of CAG-hACE2. After 2 dpi, the weight of the spleen and the area of the white pulp decreased rapidly. The increase in spleen weight can be attributed to immune response during early viral infection [41]. As viral infection progresses, depletion of B and T cells causes a decrease in spleen weight [25, 27, 41]. The pathological changes in all groups, except for the VL group of the CAG-hACE2, differed depending on the time elapsed. The rapid decrease in white pulp was faster in K18-hACE2 than in CAG-hACE2, indicating that K18-hACE2 is more susceptible to SARS-CoV-2. The VL group of CAG-hACE2 exhibited reduced white pulp atrophy at 7dpi compared with the other groups. This result indicated that CAG-hACE2 is more resistant to SARS-CoV-2 infection than the K18-hACE2 lineage. Hyperplasia of goblet cells with villi atrophy was observed in the small intestine of both lineages. This result is consistent with results of previous studies showing that the increasing number of goblet cells in COPD patients reproduced severe SARS-CoV-2 infection. Upregulation of ACE2 in goblet cells is associated with a severe infection of SARS-CoV-2 in COPD patients [42–44]. Based on these studies, hyperplasia of goblet cells can be judged as an indicator of SARS-CoV-2 infection. However, SARS-CoV-2 virus replication in the small intestine wasn't detected in the qPCR assay. For this reason, further research is needed on the effect of SARS-CoV-2 on the small intestine. Taken these results together, our study indicates that K18-hACE2 is more susceptible to SARS-CoV-2 than CAG-hACE2. These results suggest that the response to SARS-CoV-2 infection differs depending on the transgenic mouse lineage.

Based on these results, differences in the promoter used caused variations in susceptibilities of the two transgenic mouse lineages to SARS-CoV-2. However, according to a study comparing the difference in promoter expression, the expression level of the transgene under the CAG promoter was higher than that under the K18 promoter in both cells and animals [45, 46]. Contrary to the results of these previous studies, high expression of hACE2 protein was confirmed in multiple organs by western blotting in both lineages used in this study, independent of clinical symptoms. K18-hACE2 mice may be more susceptible to SARS-CoV-2 than CAG-hACE2 because the K18 promoter is an epithelial cell-derived promoter, whereas the CAG promoter is a synthetic promoter that combines cytomegalovirus enhancer with chicken beta-actin gene promoter. As an epithelial cell-derived promoter, the K18 promoter is thought to affect hACE2 expression and SARS-CoV-2 infection mechanisms. Little is known about the correlation between the K18 promoter and SARS-CoV-2 infection mechanism. Therefore, further studies are needed to identify the correlation between the K18 promoter and SARs-CoV-2 infection mechanism.

## Conclusion

In this study, we demonstrated that two different lineages of FVB background transgenic mice expressing hACE2 exhibit different clinical manifestations of SARS-CoV-2 infection,

depending on the promoter used. Based on these lineage-specific sensitivities, we offer that K18-hACE2 mouse, which shows high mortality to SARS-CoV-2 infection, is suitable for highly susceptible model of SARS-CoV-2 infection, and CAG-hACE2 mice, which shows morbidity to SARS-CoV-2 without lethality, is suitable for mild susceptible model of SARS-CoV-2 infection.

## Supporting information

**S1 Raw images.**
(PDF)

## Acknowledgments

The authors thank Han-Bi Jeong at Konkuk University for her assistance.

## Author Contributions

**Conceptualization:** Ki Taek Nam, Ho Lee, Je Kyung Seong, Yang-Kyu Choi.

**Data curation:** Sun-Min Seo, Jae Hyung Son, Hyun Ah Noh, Jun-Won Yun, Jun-Young Seo.

**Formal analysis:** Jun Won Park, Kang-Seuk Choi, Ho-Young Lee, Jeon-Soo Shin, Ki Taek Nam.

**Funding acquisition:** Je Kyung Seong, Yang-Kyu Choi.

**Investigation:** Sun-Min Seo, Jae Hyung Son, Ji-Hun Lee, Na-Won Kim, Eun-Seon Yoo, Ah-Reum Kang, Ji Yun Jang, Da In On.

**Methodology:** Sun-Min Seo, Ji-Hun Lee, Na-Won Kim, Eun-Seon Yoo, Ah-Reum Kang, Ji Yun Jang, Da In On, Jun Won Park.

**Project administration:** Hyun Ah Noh, Jun-Won Yun, Kang-Seuk Choi, Ho-Young Lee, Jeon-Soo Shin, Jun-Young Seo.

**Supervision:** Ho Lee, Je Kyung Seong, Yang-Kyu Choi.

**Writing – original draft:** Sun-Min Seo.

**Writing – review & editing:** Ho Lee, Je Kyung Seong, Yang-Kyu Choi.

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
