## [Decision Letter · Decision Letter 0]

18 Apr 2022

PONE-D-22-06945Development of transgenic models susceptible and resistant to SARS-CoV-2 infection in FVB background micePLOS ONE

Dear Dr. Choi,

Thank you for submitting your manuscript to PLOS ONE. After careful consideration, we feel that it has merit but does not fully meet PLOS ONE’s publication criteria as it currently stands. Therefore, we invite you to submit a revised version of the manuscript that addresses the points raised during the review process.

 In particular, viral replication in brain, spleen and intestine should be examined so that it can be appreciated how the new transgenic models reported herein can be useful for SARS-CoV-2 research and preclinical testing, as compared with their counterparts in B6 background reported in literature. Also, the manuscript will benefit from more discussions on the rationale of developing hACE2 transgenic mice in FVB background and the implications of your data in understanding COVID pathogenesis.

We look forward to receiving your revised manuscript.

Kind regards,

Kui Li

Academic Editor

PLOS ONE

Journal Requirements:

3. Please remove your figures from within your manuscript file, leaving only the individual TIFF/EPS image files, uploaded separately.  These will be automatically included in the reviewers’ PDF.

Reviewers' comments:

Reviewer's Responses to Questions

**Comments to the Author**

1. Is the manuscript technically sound, and do the data support the conclusions?

Reviewer #1: Partly

Reviewer #2: Partly

2. Has the statistical analysis been performed appropriately and rigorously? 

Reviewer #1: No

Reviewer #2: Yes

3. Have the authors made all data underlying the findings in their manuscript fully available?

Reviewer #1: Yes

Reviewer #2: Yes

4. Is the manuscript presented in an intelligible fashion and written in standard English?

Reviewer #1: Yes

Reviewer #2: Yes

5. Review Comments to the Author

Reviewer #1: In this study, Seo et al. compared two transgenic mouse models for their susceptibility to SARS-CoV-2 infection. These two transgenic mice models are FVB background and express human ACE2 driven by two different promoters (K18 and CAG). They found that K18-hACE2 FVB mice were more susceptible to SARS-CoV-2 infection than the CAG-hACE2 FVB mice. Overall, the subject of this research is interesting as hACE2-transgenic mice have been broadly used in SARS-CoV and SARS-CoV-2 research, but no clear comparative data has been published regarding the mice expressing hACE2 driven by different promoters. This study tried to fill this gap and found different results compared to a previous model (Masamitsu et al., 2021, JCI insight). Nonetheless, this study lacks rigorous analyses of the two models and does not comprehensively discuss them with previous models.

General comments:

1. As we’ve already known, the genetic background of mice is very important for the susceptibility to many virus infections, including SARS-CoV and SARS-CoV-2. Early studies have shown the utility of K18 and CAG mice (B6 background) for SARS-CoV-2 research, this study shows that the CAG-hACE2 transgenic FVB mice may not be a good model for SARS-CoV-2. The authors created the FVB transgenic mice but did not say what FVB mice were and why to use FVB mice.

2. Masamitsu et al. (2021, JCI insight) reported the CAG promoter-driven hACE2 mouse model was highly susceptible to SARS-CoV-2. These mice, infected via intranasal or intratracheal route, exhibited severe disease. Since the K18-hACE2 FVB mice were highly susceptible to SARS-CoV-2, what would be the potential reasons for the low susceptibility of the CAG-hACE2 mice?

3. The authors briefly describe what the K18 promotor is but do not say anything about the CAG promotor. If the authors want to compare the utility of two transgenic mice, they should discuss the differences between the two promotors and the potential expression distribution of the targeted protein (human ACE2), and what effects would be resulted from the different protein distribution.

4. If the authors want to compare the susceptibility of two transgenic mouse models, they should compare these mice infected with the same dose of virus. It is not very meaningful to compare the low- and high-dose groups within the same transgenic mice as the differences would be expected.

5. If the spleen and intestine respond to SARS-CoV-2 infection, viral titers or RNA levels in these organs should be investigated as well as the inflammatory response (e.g. cytokine levels).

Specific comments:

1. Fig 1: First, according to the beta-actin levels, would be the undetectable ACE2 in the liver due to inadequate total protein loaded? The same amount of total protein should be analyzed. An alternative approach is to detect by qPCR. Second, the brain and spleen should be included as 1) early studies showed that the brain of the K18-hACE2 mice is targeted by SARS-CoV-2, and 2) later data include the spleen. Third, later data no mock control for the IHC data.

2. Fig 2: Specify the statistical tests in this and other figure legends.

3. Fig 3: The viral RNA levels in the lungs, spleens, and intestines should be determined as later data include these organs. For the low-dose group, any explanation for no viral propagation during the course of infection? Is it meaningful to calculate the p values between the low and high-dose groups?

4. Fig 7: First, what is the white pulp percentage, the number of pulp per tissue slide, or the area of the pulp? If it was the number, how many slides were counted? If it was the area, how many pulps were counted? Second, for the number of goblet cells, how many villi were counted? Third, goblet cells cannot be seen in the current images. Enlarged images should be provided. Fourth, the viral loads in the spleen and intestine should be determined.

Reviewer #2: In the manuscript titled “Development of transgenic models susceptible and resistant to SARS-CoV-2 infection in FVB background mice,” Seo SM and colleagues compare host response and pathology of K18-hACE2 mouse model of SARS-CoV-2 infection to another transgenic line CAG-hACE2 mice both on FVB background. The authors find similarities and differences in SARS-CoV-2 replication and pathology in these strains. Specifically, SARS-CoV-2 replicated to high titers and caused severe clinical illness, fatal pneumonia, and mortality in K18 mice compared to CAG-hACE2 mice. Based on these results, the authors conclude that the K18-hACE2 mice are useful for vaccine testing while CAG-hACE2 mice for therapeutic evaluation. Specific comments are listed below.

Major comments:

1) Given the availability of K18-hACE2 mice on B6 background for COVID19 studies, which is a robust model, it is unclear why the authors chose to develop K18-hACE2 and CAG-hACE2 mice on an FVB background.

2) K18-hACE2 mice on B6 background develop fatal brain infection, which largely contributes to morbidity and mortality observed in these mice. However, the authors did not assess brain infection in mice on FVB background. If mice on FVB background do not show brain infection, then perhaps these mice would be better models to study SARS-CoV-2 infection and lung pathology.

3) The authors conclude that K18-hACE2 mice are better for vaccine testing and CAG-hACE2 mice for therapeutic evaluation. However, no rationale or basis is provided to support these conclusions.

4) CAG-hACE2 mice infected with a high dose of the virus show similar or high lung titers compared to low dose virus infection in K18 mice (Figs 2 and 3). Yet, high dose CAG-hACE2 mice do not show signs of morbidity or pathology observed in low dose K18 mice. The authors should discuss the basis for these differences.

5) K18-hACE2 mice have high ACE2 expression in other tissues compared to CAG-hACE2. In correlation, increased pathology is observed in the spleen and intestines in K18 mice. Do these changes correlate with virus infection these tissues, or the changes observed in non-lung tissues are due to inflammation?

6) Discussion should include the implications of the changes observed in mouse lung and non-lung tissues and how these changes correlate with changes in lungs and other tissues of patients with severe and mild-moderate covid19.

Minor comments:

1) Line 65: Change the sentence to mean “spread of SARS-CoV-2,” not “spread of COVID19,” as it is the virus that spreads.

2) Line 206: The virus dose provided here is different from the methods section. Please correct.

3) Line 371-372: Unclear what and how these studies suggest a new line of research.

4) The authros make several claims not supported by the results. This reviewer suggests authors to carefully read the manuscript and modify the statements to reflect the results.

6. PLOS authors have the option to publish the peer review history of their article (what does this mean?). If published, this will include your full peer review and any attached files.

Reviewer #1: No

Reviewer #2: No

---

## [Author Response · Author response to Decision Letter 0]

1 Jun 2022

Kui Li, Ph.D

Academic Editor

PLOS ONE

Dear Dr. Li 

Subject: Development of transgenic models susceptible and resistant to SARS-CoV-2 infection in FVB background mice [PONE-D-22-06945]

Thank you for inviting us to submit a revised draft of our manuscript entitled, "Development of transgenic models susceptible and resistant to SARS-CoV-2 infection in FVB background mice" to PLOS ONE. We also appreciate the time and effort you and each of the reviewers have dedicated to providing insightful feedback on ways to strengthen our paper. Thus, it is with great pleasure that we resubmit our article for further consideration. We have incorporated changes that reflect the detailed suggestions you have graciously provided. We also hope that our edits and the responses we provide below satisfactorily address all the issues and concerns you and the reviewers have noted.

To facilitate your review of our revisions, the following is a point-by-point response to the questions and comments.

Proposals from Academic Editor

Answer: We revised our manuscript based on the PLOS ONE style template.

2. PLOS ONE now requires that authors provide the original uncropped and unadjusted images underlying all blot or gel results reported in a submission’s figures or Supporting Information files. This policy and the journal’s other requirements for blot/gel reporting and figure preparation are described in detail at https://journals.plos.org/plosone/s/figures#loc-blot-and-gel-reporting-requirements and https://journals.plos.org/plosone/s/figures#loc-preparing-figures-from-image-files. When you submit your revised manuscript, please ensure that your figures adhere fully to these guidelines and provide the original underlying images for all blot or gel data reported in your submission. See the following link for instructions on providing the original image data: https://journals.plos.org/plosone/s/figures#loc-original-images-for-blots-and-gels. In your cover letter, please note whether your blot/gel image data are in Supporting Information or posted at a public data repository, provide the repository URL if relevant, and provide specific details as to which raw blot/gel images, if any, are not available. Email us at plosone@plos.org if you have any questions.

Answer: We added original blot image in our manuscript based on the PLOS ONE policy.

3. Please remove your figures from within your manuscript file, leaving only the individual TIFF/EPS image files, uploaded separately. These will be automatically included in the reviewers’ PDF.

Answer: We revised our manuscript by removing the figure image from the manuscript and uploading it individually.

General comments from Reviewer #1:

In this study, Seo et al. compared two transgenic mouse models for their susceptibility to SARS-CoV-2 infection. These two transgenic mice models are FVB background and express human ACE2 driven by two different promoters (K18 and CAG). They found that K18-hACE2 FVB mice were more susceptible to SARS-CoV-2 infection than the CAG-hACE2 FVB mice. Overall, the subject of this research is interesting as hACE2-transgenic mice have been broadly used in SARS-CoV and SARS-CoV-2 research, but no clear comparative data has been published regarding the mice expressing hACE2 driven by different promoters. This study tried to fill this gap and found different results compared to a previous model (Masamitsu et al., 2021, JCI insight). Nonetheless, this study lacks rigorous analyses of the two models and does not comprehensively discuss them with previous models.

1. As we’ve already known, the genetic background of mice is very important for the susceptibility to many virus infections, including SARS-CoV and SARS-CoV-2. Early studies have shown the utility of K18 and CAG mice (B6 background) for SARS-CoV-2 research, this study shows that the CAG-hACE2 transgenic FVB mice may not be a good model for SARS-CoV-2. The authors created the FVB transgenic mice but did not say what FVB mice were and why to use FVB mice.

Answer: Thank you for providing these insights. You have raised an important point; however, we believe that CAG-hACE2 transgenic FVB mouse is a good model for SARS-CoV-2 because CAG-hACE2 transgenic FVB mouse showed morbidity without lethality. We confirmed through additional research that these results in CAG-hACE2 due to the absence of viral replication in CNS, unlike K18-hACE2. For this reason, CAG-hACE2 transgenic FVB mice can be considered as a SARS-CoV-2 model for research on lung pathology. For this statement we have included revised Fig. 3 (p. 11), additional results in (p. 11, line 258-269), and discussion in (p. 17, line 417-422 with reference [40]) and marked it with red color.

Also, we added (p. 16, lines 390-393) with reference [33-35] and marked it with red color.

2. Masamitsu et al. (2021, JCI insight) reported the CAG promoter-driven hACE2 mouse model was highly susceptible to SARS-CoV-2. These mice, infected via intranasal or intratracheal route, exhibited severe disease. Since the K18-hACE2 FVB mice were highly susceptible to SARS-CoV-2, what would be the potential reasons for the low susceptibility of the CAG-hACE2 mice?

Answer: We revised (p. 16, lines 405-408) and added reference [15] using red color.

3. The authors briefly describe what the K18 promotor is but do not say anything about the CAG promotor. If the authors want to compare the utility of two transgenic mice, they should discuss the differences between the two promotors and the potential expression distribution of the targeted protein (human ACE2), and what effects would be resulted from the different protein distribution.

Answer: We have supplemented the brief describe on the CAG promoter in (p. 16, lines 405-408) and (p.18, lines 449-450) and marked in red color.

In both K18-hACE2 and CAG-hACE2, the hACE2 expression level in the lung, the main target organ of the SARs-CoV-2, was good. However, the K18 promoter, epithelial cell derived promoter, was intensively expressed in the lung, and the CAG promoter, a synthetic promoter that combines cytomegalovirus enhancer with chicken beta-actin gene promoter, was evenly expressed in all organs. These results exhibit a similar pattern to the C57BL/6 background hACE2 TG, described in the reference [15,16]. Cause the K18-hACE2 was generated with epithelial cell derived promoter, the expression of hACE2 in the respiratory epithelium was biased, and this is expected to result in severe pneumonia compared to CAG-hACE2.

4. If the authors want to compare the susceptibility of two transgenic mouse models, they should compare these mice infected with the same dose of virus. It is not very meaningful to compare the low- and high-dose groups within the same transgenic mice as the differences would be expected.

Answer: We agree with your suggestion that comparing with the same dose of virus is needed. So, we have added statistical analysis results in (p. 10, lines 226-227), (p. 10, lines 233-234), (p. 11, lines 256-257), (p. 11, lines 261-263), (p. 11, lines 266-267), (p. 12, lines 291-293), (p. 12-13, lines 302-304), (p. 13, lines 328-329), (p. 14, lines 342-343), and (p. 14, lines 352-354). Added sentences marked with red color. However, we have retained some of our arguments because, in establishing the novel model for SARS-CoV-2 infection, we thought it was important to compare the differences in pathogenicity depending on administered dose.

5. If the spleen and intestine respond to SARS-CoV-2 infection, viral titers or RNA levels in these organs should be investigated as well as the inflammatory response (e.g. cytokine levels).

Answer: We performed qPCR assay for duodenum and these results and discussion added in Fig. 3(p. 11), additional results in (p. 11, lines 267-269), and revised discussion in (p. 17-18, lines 433-439) (Added and revised sentences marked with red color). However, we couldn’t perform qPCR assay for spleen because entire spleen was formalin fixed for histopathological analysis to analysis white pulp atrophy.

Specific comments from Reviewer #1:

1. Fig 1: First, according to the beta-actin levels, would be the undetectable ACE2 in the liver due to inadequate total protein loaded? The same amount of total protein should be analyzed. An alternative approach is to detect by qPCR. Second, the brain and spleen should be included as 1) early studies showed that the brain of the K18-hACE2 mice is targeted by SARS-CoV-2, and 2) later data include the spleen. Third, later data no mock control for the IHC data.

Answer: Thank you for your suggestion. As you see in the original blot image of the supplemental figure, the CAG-hACE2 blot image used in Fig 1 was mouse #4. In the original blot image of mouse #3, hACE2 was expressed in all organs including the liver. However, we presented the blot image of mouse #4 by focusing on hACE2 expression in the lung, the main target of SARS-CoV-2. In the case of K18-hACE2, hACE2 expression level in the organs including the liver, heart, and intestine was very weak or absent, compared to CAG-hACE2, despite loading the same amount of total protein. 

We agree with that brain and spleen is important target for the SARS-CoV-2. We have included additional research on the brain. Those results are available in revised Fig. 3 (p. 11), additional results in (p. 11, line 258-267), and discussion in (p. 17, line 417-422 with reference [40]) (Added and revised sentences marked with red color). However, additional research of the spleen was not possible, please see general comments 5 above.

We agree with mock control for the IHC data is necessary. We added mock control in Fig 1 and marked with red color.

2. Fig 2: Specify the statistical tests in this and other figure legends.

Answer: We revised Fig 2 (p. 10, lines 240-241), Fig 3 (p. 12, line 278), Fig 4 (p. 13, line 313), Fig 6 (p. 15, line 366), and Fig 8 (p. 15, line 381) and marked it with red color.

3. Fig 3: The viral RNA levels in the lungs, spleens, and intestines should be determined as later data include these organs. For the low-dose group, any explanation for no viral propagation during the course of infection? Is it meaningful to calculate the p values between the low and high-dose groups?

Answer: Thank you for your suggestion. In low-dose group, although the viral load maintained similar level from 2 dpi to 7 dpi (Fig 3), histopathological findings showed more severe lesions depending on the time passed (Fig 5). Based on these results, we thought that virus propagation appeared in the low-dose group. 

For the viral RNA levels in spleens and intestines, please see general comments 5 above. Also, the meaning of p values between the low- and high-dose group, please see general comments 4 above

4. Fig 7: First, what is the white pulp percentage, the number of pulp per tissue slide, or the area of the pulp? If it was the number, how many slides were counted? If it was the area, how many pulps were counted? Second, for the number of goblet cells, how many villi were counted? Third, goblet cells cannot be seen in the current images. Enlarged images should be provided. Fourth, the viral loads in the spleen and intestine should be determined.

Answer: We calculated white pulp percentage by the area of white pulp relative to the area of the entire spleen with (magnification x100). The criteria for the goblet cells are the average of goblet cells from at least 10 villi of jejunum (Table 2). Also, we revised Fig 7 and input the enlarged image of the small intestine. The viral loads of the spleen and intestine, please see general comments 5 above.

Major comments from Reviewer #2:

In the manuscript titled “Development of transgenic models susceptible and resistant to SARS-CoV-2 infection in FVB background mice,” Seo SM and colleagues compare host response and pathology of K18-hACE2 mouse model of SARS-CoV-2 infection to another transgenic line CAG-hACE2 mice both on FVB background. The authors find similarities and differences in SARS-CoV-2 replication and pathology in these strains. Specifically, SARS-CoV-2 replicated to high titers and caused severe clinical illness, fatal pneumonia, and mortality in K18 mice compared to CAG-hACE2 mice. Based on these results, the authors conclude that the K18-hACE2 mice are useful for vaccine testing while CAG-hACE2 mice for therapeutic evaluation. Specific comments are listed below. 

1. Given the availability of K18-hACE2 mice on B6 background for COVID19 studies, which is a robust model, it is unclear why the authors chose to develop K18-hACE2 and CAG-hACE2 mice on an FVB background.

Answer: Thank you for your suggestion. We have reflected this comment by (p. 16, lines 390-393) with reference [33-35] and marked it with red color.

2. K18-hACE2 mice on B6 background develop fatal brain infection, which largely contributes to morbidity and mortality observed in these mice. However, the authors did not assess brain infection in mice on FVB background. If mice on FVB background do not show brain infection, then perhaps these mice would be better models to study SARS-CoV-2 infection and lung pathology.

Answer: You have raised an important question. We agree with you and have incorporated this suggestion throughout our paper. We have included revised Fig. 3 (p. 11), additional results in (p. 11, line 258-269), and discussion in (p. 17, line 417-422 with reference [40]) and marked it with red color.

3. The authors conclude that K18-hACE2 mice are better for vaccine testing and CAG-hACE2 mice for therapeutic evaluation. However, no rationale or basis is provided to support these conclusions. 

Answer: We agree that our conclusions were lack of rationale. So, we have clarified our conclusion with revised sentence in (p. 18, lines 459-462) and marked with red color.

4. CAG-hACE2 mice infected with a high dose of the virus show similar or high lung titers compared to low dose virus infection in K18 mice (Figs 2 and 3). Yet, high dose CAG-hACE2 mice do not show signs of morbidity or pathology observed in low dose K18 mice. The authors should discuss the basis for these differences.

Answer: Thank you for your suggestion. We have incorporated your comments by (p. 17, line 417-422 with reference [40]) and marked it with red color.

5. K18-hACE2 mice have high ACE2 expression in other tissues compared to CAG-hACE2. In correlation, increased pathology is observed in the spleen and intestines in K18 mice. Do these changes correlate with virus infection these tissues, or the changes observed in non-lung tissues are due to inflammation?

Answer: Thank you for you suggestion. White pulp atrophy in the spleen due to SARS-CoV-2 infection has been reported in few case reports in human. We mentioned white pulp atrophy in (p. 13, lines 329-330 with references [25-27]) marked with red color. The hyperplasia of goblet cells in the small intestine was first reported in our study. Contrary to previous studies, which detected the virus in the small intestine, the virus was undetected in our qPCR assay. We considered that a very little amount of the virus couldn't be detected because below the limit of detection of the standard curve we drew.

6. Discussion should include the implications of the changes observed in mouse lung and non-lung tissues and how these changes correlate with changes in lungs and other tissues of patients with severe and mild-moderate covid19. 

Answer: We have incorporated your comments by (p. 17, lines 413-414, with references [36-39]), and (p. 13, lines 329-330 with references [25-27]) and marked with red color. SARS-CoV-2 infection in patient’s small intestine was previously reported, however, the goblet cell hyperplasia in the small intestine is first reported in our research.

Minor comments from Reviewer #2:

1. Line 65: Change the sentence to mean “spread of SARS-CoV-2,” not “spread of COVID19,” as it is the virus that spreads. 

Answer: We revised word COVID19 to SARS-CoV-2 in (p. 3, line 63) and marked it with red color.

2. Line 206: The virus dose provided here is different from the methods section. Please correct.

Answer: We revised (p. 9, line 209) and marked it with red color.

3. Line 371-372: Unclear what and how these studies suggest a new line of research.

Answer: We agree with you. We revised word research to model in (p. 16, line 395) and marked it with red color.

4. The authors make several claims not supported by the results. This reviewer suggests authors to carefully read the manuscript and modify the statements to reflect the results.

Answer: We have corrected and supplemented the insufficient statements by referring to the revisions sent by two reviewers.

Again, thank you for giving us the opportunity to strengthen our manuscript with your valuable comments and queries. We have worked hard to incorporate your feedback and hope that these revisions persuade you to accept our submission.

Sincerely yours,

Yang-Kyu Choi

---

## [Decision Letter · Decision Letter 1]

21 Jun 2022

PONE-D-22-06945R1Development of transgenic models susceptible and resistant to SARS-CoV-2 infection in FVB background micePLOS ONE

Dear Dr. Choi,

Thank you for submitting your manuscript to PLOS ONE. After careful consideration, we feel that it has merit but does not fully meet PLOS ONE’s publication criteria as it currently stands. Therefore, we invite you to submit a revised version of the manuscript that addresses the points raised during the review process.

 While the manuscript is substantially improved, some minor issues remain. As the reviewers pointed out, some corrections and additional clarifications are needed. Please ensure that your decision is justified on PLOS ONE’s publication criteria and not, for example, on novelty or perceived impact.

We look forward to receiving your revised manuscript.

Kind regards,

Kui Li

Academic Editor

PLOS ONE

Journal Requirements:

Additional Editor Comments (if provided):

While the revised manuscript is substantially improved, some minor issues remain. As you see from the reviewers' comments, some corrections and additional clarifications are needed.

Reviewers' comments:

Reviewer's Responses to Questions

**Comments to the Author**

1. If the authors have adequately addressed your comments raised in a previous round of review and you feel that this manuscript is now acceptable for publication, you may indicate that here to bypass the “Comments to the Author” section, enter your conflict of interest statement in the “Confidential to Editor” section, and submit your "Accept" recommendation.

Reviewer #1: All comments have been addressed

Reviewer #2: All comments have been addressed

2. Is the manuscript technically sound, and do the data support the conclusions?

Reviewer #1: Yes

Reviewer #2: Partly

3. Has the statistical analysis been performed appropriately and rigorously? 

Reviewer #1: Yes

Reviewer #2: Yes

4. Have the authors made all data underlying the findings in their manuscript fully available?

Reviewer #1: Yes

Reviewer #2: Yes

5. Is the manuscript presented in an intelligible fashion and written in standard English?

Reviewer #1: Yes

Reviewer #2: Yes

6. Review Comments to the Author

Reviewer #1: Overall, the authors properly addressed the reviewers’ comments. But oddly, the authors revised figure 1 and switched the labeling of Fig1a and 1b. The revised fig.1 indicated that CAG-hACE2 mice organs (kidney, lung, heart, intestine) expressed high levels of hACE2, whereas the lungs of the k18-hACE2 mice were the primary organ expressing hACE2. This contradicts the rest of the data (fig3, 7, 8) and statement (lines 265-276) as well as previous studies. Please examine.

Reviewer #2: The authors have made significant revision and addressed majority of the concerns. There are several minor errors that need to be corrected. Comments are listed below.

1) At several instances where a new text is added, the authors mention " a significant difference was observed" or used similar language throughout the text. This should be corrected to clearly mention what differences were observed, and whether the difference was increased or decreased between the groups. Also- include the significance of these differences.

2) The authors use the word " virus propagation" in certain tissue on numerous occasions. This should be corrected to virus titers or virus replication or virus load.

3) In the earlier iteration, this reviewer asked authors to provide justification for recommending use of K18-hACE2 mice for vaccine evaluation versus CAG-hACE2 mice for therapeutic assessment. However, this statement is still not well justified. Therefore, should be removed from the text or provide better explanation.

7. PLOS authors have the option to publish the peer review history of their article (what does this mean?). If published, this will include your full peer review and any attached files.

Reviewer #1: No

Reviewer #2: No

---

## [Author Response · Author response to Decision Letter 1]

4 Jul 2022

Dear Dr. Li 

Subject: Development of transgenic models susceptible and resistant to SARS-CoV-2 infection in FVB background mice [PONE-D-22-06945R1]

Thank you for inviting us to submit a revised draft of our manuscript entitled, "Development of transgenic models susceptible and resistant to SARS-CoV-2 infection in FVB background mice" to PLOS ONE. We also appreciate the time and effort you and each of the reviewers have dedicated to providing insightful feedback on ways to strengthen our paper. Thus, it is with great pleasure that we resubmit our article for further consideration. We have incorporated changes that reflect the detailed suggestions you have graciously provided. We also hope that our edits and the responses we provide below satisfactorily address all the issues and concerns you and the reviewers have noted.

Proposals from Academic Editor

Thank you for submitting your manuscript to PLOS ONE. After careful consideration, we feel that it has merit but does not fully meet PLOS ONE’s publication criteria as it currently stands. Therefore, we invite you to submit a revised version of the manuscript that addresses the points raised during the review process.

While the manuscript is substantially improved, some minor issues remain. As the reviewers pointed out, some corrections and additional clarifications are needed

1. Please review your reference list to ensure that it is complete and correct. If you have cited papers that have been retracted, please include the rationale for doing so in the manuscript text or remove these references and replace them with relevant current references. Any changes to the reference list should be mentioned in the rebuttal letter that accompanies your revised manuscript. If you need to cite a retracted article, indicate the article’s retracted status in the References list and also include a citation and full reference for the retraction notice.

Answer: We revised our references based on “Vancouver” style in (p. 20, lines 497, 504, 507, and 509), (p. 21, lines 519, and 537), (p. 22, lines 545, 547, 549-550, 556, 558, and 564), (p. 23, lines 571), and (p. 24, line 599). We marked the revised sentences with red color.

2. While the revised manuscript is substantially improved, some minor issues remain. As you see from the reviewers' comments, some corrections and additional clarifications are needed.

Answer: We revised our manuscript based on reviewers’ comments.

Review comments from Reviewer #1:

1. Overall, the authors properly addressed the reviewers’ comments. But oddly, the authors revised figure 1 and switched the labeling of Fig1a and 1b. The revised fig.1 indicated that CAG-hACE2 mice organs (kidney, lung, heart, intestine) expressed high levels of hACE2, whereas the lungs of the k18-hACE2 mice were the primary organ expressing hACE2. This contradicts the rest of the data (fig3, 7, 8) and statement (lines 265-276) as well as previous studies. Please examine.

Answer: Thank you for providing these insights. In our original manuscript, there was an error in the labeling of Fig1a and 1b. So, we switched the labeling in the revised manuscript. In contrast to qPCR and histopathology performed in the small intestine, Western blot was performed in the large intestine. For this reason, it is considered that a contradiction appeared between the expression level of hACE2 and pathological scoring. We revised our manuscript (p. 8, line 190, marked with red color.), Fig 1. (from intestine to large intestine), and supplemental figure 1. (from intestine to large intestine).

Review comments from Reviewer #2:

The authors have made significant revision and addressed majority of the concerns. There are several minor errors that need to be corrected. Comments are listed below.

1. At several instances where a new text is added, the authors mention " a significant difference was observed" or used similar language throughout the text. This should be corrected to clearly mention what differences were observed, and whether the difference was increased or decreased between the groups. Also- include the significance of these differences.

Answer: Thank you for your suggestion. We have reflected this comment by (p. 10, lines 226-227), (p. 10, lines 233-234), (p. 11, lines 256-257), (p. 11, lines 262-263), (p. 11, lines 266-268), (p. 12, lines 291-293), (p. 13, lines 302-304), (p. 14, lines 328-329), (p. 14, lines 342-343), and (p. 14, lines 352-354). Revised and added sentences marked with red color.

2. The authors use the word " virus propagation" in certain tissue on numerous occasions. This should be corrected to virus titers or virus replication or virus load.

Answer: We agree with your assessment. We revised the word “virus propagation” to “viral replication” and marked it with red color in (p. 11, line 258, 264, and 268), and (p.17, line 416).

3. In the earlier iteration, this reviewer asked authors to provide justification for recommending use of K18-hACE2 mice for vaccine evaluation versus CAG-hACE2 mice for therapeutic assessment. However, this statement is still not well justified. Therefore, should be removed from the text or provide better explanation.

Answer: We agree that our conclusions lack proper explanation. So, we have clarified our conclusion with a revised sentence (p. 2, lines 46-47) (p. 18, lines 457-460) and marked it with red color.

Again, thank you for giving us the opportunity to strengthen our manuscript with your valuable comments and queries. We have worked hard to incorporate your feedback and hope that these revisions persuade you to accept our submission.

Sincerely yours,

---

## [Decision Letter · Decision Letter 2]

12 Jul 2022

Development of transgenic models susceptible and resistant to SARS-CoV-2 infection in FVB background mice

PONE-D-22-06945R2

Dear Dr. Choi,

We’re pleased to inform you that your manuscript has been judged scientifically suitable for publication and will be formally accepted for publication once it meets all outstanding technical requirements.

Kind regards,

Kui Li

Academic Editor

PLOS ONE

Additional Editor Comments (optional):

Reviewers' comments:

Reviewer's Responses to Questions

**Comments to the Author**

1. If the authors have adequately addressed your comments raised in a previous round of review and you feel that this manuscript is now acceptable for publication, you may indicate that here to bypass the “Comments to the Author” section, enter your conflict of interest statement in the “Confidential to Editor” section, and submit your "Accept" recommendation.

Reviewer #1: All comments have been addressed

Reviewer #2: All comments have been addressed

2. Is the manuscript technically sound, and do the data support the conclusions?

Reviewer #1: Yes

Reviewer #2: Yes

3. Has the statistical analysis been performed appropriately and rigorously? 

Reviewer #1: Yes

Reviewer #2: Yes

4. Have the authors made all data underlying the findings in their manuscript fully available?

Reviewer #1: Yes

Reviewer #2: Yes

5. Is the manuscript presented in an intelligible fashion and written in standard English?

Reviewer #1: Yes

Reviewer #2: Yes

6. Review Comments to the Author

Reviewer #1: (No Response)

Reviewer #2: The authors have addressed all the remaining concerns and the manuscript is acceptable for publication.

7. PLOS authors have the option to publish the peer review history of their article (what does this mean?). If published, this will include your full peer review and any attached files.

Reviewer #1: No

Reviewer #2: No

---

## [Editor Report · Acceptance letter]

15 Jul 2022

PONE-D-22-06945R2 

Development of transgenic models susceptible and resistant to SARS-CoV-2 infection in FVB background mice 

Dear Dr. Choi:

I'm pleased to inform you that your manuscript has been deemed suitable for publication in PLOS ONE. Congratulations! Your manuscript is now with our production department. 

Kind regards, 

on behalf of

Dr. Kui Li 

Academic Editor

PLOS ONE